# Assessing Spatial Distribution of Multicellular Self-Assembly Enables the Prediction of Phenotypic Heterogeneity in Glioblastoma

**DOI:** 10.3390/cancers14235910

**Published:** 2022-11-30

**Authors:** Junghwa Cha, Woogwang Sim, Insung Yong, Junseong Park, Jin-Kyoung Shim, Jong Hee Chang, Seok-Gu Kang, Pilnam Kim

**Affiliations:** 1Department of Bio and Brain Engineering, Korea Advanced Institute of Science and Technology (KAIST), Daejeon 34141, Republic of Korea; 2KAIST Institute for Health Science and Technology, Daejeon 34141, Republic of Korea; 3Department of Bioengineering, University of California, Berkeley, CA 94720, USA; 4Department of Anatomy, University of California, San Francisco, CA 94143, USA; 5Department of Neurosurgery, Severance Hospital, Yonsei University College of Medicine, Seoul 03722, Republic of Korea; 6Precision Medicine Research Center, College of Medicine, The Catholic University of Korea, Seoul 06591, Republic of Korea; 7Department of Medical Science, Yonsei University Graduate School, Seoul 03722, Republic of Korea

**Keywords:** multicellular self-assembly, tumor heterogeneity, glioblastomas, heterotypic cultivation, prognosis prediction

## Abstract

**Simple Summary:**

The heterogeneity of tumors is one of the primary obstacles to successful treatment. Although the examination of tumor heterogeneity is essential for comprehending tumor characteristics and planning therapeutic strategies, it remains challenging to assess and evaluate heterogeneous tumor populations. Here, we propose the self-assembly-based evaluation method, which is capable of predicting inter/intracellular heterogeneity in glioblastoma. Depending on their self-assembly pattern, heterotypic multicellular aggregates (hMA) are formed by mixed populations of glioblastoma cells. The cells located at the outermost hMA exhibit a diminished temozolomide response, and are related with poor patient survival. Our findings imply that the multicellular self-assembly pattern is indicative of the intertumoral and intra-patient heterogeneity of glioblastomas, and is also prognostic of the therapeutic response.

**Abstract:**

Phenotypic heterogeneity of glioblastomas is a leading determinant of therapeutic resistance and treatment failure. However, functional assessment of the heterogeneity of glioblastomas is lacking. We developed a self-assembly-based assessment system that predicts inter/intracellular heterogeneity and phenotype associations, such as cell proliferation, invasiveness, drug responses, and gene expression profiles. Under physical constraints for cellular interactions, mixed populations of glioblastoma cells are sorted to form a segregated architecture, depending on their preference for binding to cells of the same phenotype. Cells distributed at the periphery exhibit a reduced temozolomide (TMZ) response and are associated with poor patient survival, whereas cells in the core of the aggregates exhibit a significant response to TMZ. Our results suggest that the multicellular self-assembly pattern is indicative of the intertumoral and intra-patient heterogeneity of glioblastomas, and is predictive of the therapeutic response.

## 1. Introduction

Heterogeneity is evident within a tumor (intratumoral heterogeneity), as well as between tumors (intertumoral heterogeneity) in terms of cellular morphology, gene expression, metabolism, motility, proliferation, and metastatic potential [1]. Tumors become more heterogeneous as they progress, developing a diverse collection of cells with distinct molecular signatures that exhibit varying levels of treatment sensitivity [2]. Such heterogeneity is associated with different drug responses of tumor cell populations across and within disease sites, as well as over time [3]. Glioblastomas are characterized by marked intratumoral and inter-patient heterogeneity [4]. As such, glioblastoma heterogeneity renders the development of effective therapies challenging; additionally, accurate delineation of tumor heterogeneity is critical in planning for proper therapy.

Immunohistochemistry has been utilized as a widely applicable approach for detecting various subtypes based on their genetic properties [5,6]. Recent analytical approaches, such as bulk expression-based subtyping [7] and single cell-based analyses [1,8], have the ability to give high-resolution information on genomic, transcriptomic, and epigenetic variations within glioblastoma populations. To evaluate tumor heterogeneity, limitations are placed on selective marker-based analysis and site-dependent marker expression. Moreover, omics-based analyses are impractical in the majority of clinical situations due to experimental challenges and the underlying phenotypic plasticity of cells [9]. Context-based functional evaluation techniques, such as the cell-migration assay, have proven inter-patient heterogeneity and predicted patient-specific therapeutic outcomes [10,11] to address these limitations. In order to design therapeutic regimens for glioblastoma patients, faster assessment approaches that can predict intratumoral heterogeneity within the same tumor specimen are still required.

Cell affinity has been proposed as a notion relating to the tendency of cells to interact preferentially with cells of the same type [12]. The cells in a mixed aggregate form a well-segregated population by recognizing the identity of their neighbors and congregating with cells of the same type [13]. Cell–cell adhesion molecules (CAMs), such as homotypic cell–cell adhesion, have become one of the most important regulators for cell–cell affinity and tissue segregation [14]. Different gene expression levels of cell affinity mediate the formation of spatially discrete compartments in multicellular aggregates during the developmental process [15].

Likewise, cell affinity is highly involved during tumorigenesis; the loss of intercellular affinity in the tumor mass endows neoplastic subpopulations with an invasive phenotype. Cells within a heterogeneous population exhibit diverse associative preferences; thus, cells tend to cohere with neighbor cells, self-organizing into suitable configurations and patterns [15,16]. Consequently, we hypothesized that differential intercellular affinity could serve as a valuable phenotypic marker of intratumoral heterogeneity. We postulated, based on the notion of self-organization through variable cell–cell cohesive contacts [17,18], that the patterns of self-assembled glioblastoma cells could be a morphometric indicator for identifying cells with heterogeneous phenotypes within tumors (Figure 1).

## 2. Materials and Methods

### 2.1. Cell Culture

Tumorspheres (TSs) of pdGCs were provided by the laboratory of Dr. Seok-Gu Kang at Yonsei University of College of Medicine, and were derived as previously reported [19]. Approval was given by the Institutional Review Board of Severance Hospital, Yonsei University College of Medicine (4-2014-0649). Informed consent was provided according to the Collaborative Institutional Training Initiative (CITI) Program, as specified in the Declaration of Helsinki (K-2014-13752477).

After the specimens from human GBM patients were freshly obtained from the operation room, the pdGCs were isolated using a mechanical dissociation method. The specimens were minced and dissociated with a scalpel in DMEM/nutrient mixture F-12 (DMEM/F12), and then filtered through a series of 100-μm nylon mesh cell strainers. pdGCs (listed in Table 1) were cultured on DMEM/nutrient mixture F-12 (DMEM/F12) (Welgene, Gyeongsan, Republic of Korea) supplemented with 1% P/S (Welgene), 1 × B27 (Invitrogen, Carlsbad, CA, USA), 20 ng/mL basic fibroblast growth factor (Invitrogen), and 20 ng/mL epidermal growth factor (Invitrogen). Confluent TSs were passaged every 5 days (1:5 ratio) by dissociating with Accutase (Invitrogen). pdGCs were maintained at 37 °C in an atmosphere of 5% CO_2_ and 95% air. U87 and U251 glioblastoma cells were cultivated in Dulbecco’s modified Eagle’s medium (DMEM), supplemented with 10% fetal bovine serum and 1% penicillin/streptomycin (P/S) (Welgene, Gyeongsan, Republic of Korea). For the experiments, U87 and U251 cells were passaged every 3–5 days (1: 5 ratio) after they were dissociated using trypsin/EDTA solution (Welgene). To visualize subpopulations, cells for heterotypic culture were labeled with CellTracker™ (Invitrogen), with different fluorophores.

### 2.2. Preparation of the Multicellular Aggregation System

The multicellular aggregation system was prepared by the process of replica molding of SpheroFilm^TM^ on agarose hydrogels in order to generate a 15-by-15 array, composed of 225 microwells of 500-μm diameter and 500-μm depth, as described in our previous study [20]. SpheroFilm^TM^ was coated with Trichloro(1H, 1H, 2H, 2H-perfluorooctyl)silane (Sigma) to create an anti-stiction layer with hydrophobic properties as a master pattern. Prepolymers of polydimethylsiloxane (PDMS) (10:1 ration of *w*/*w* resin:crosslinking agent) were poured onto the silane-coated SpheroFilm^TM^ and solidified in a convection oven (70 °C, 1 h). Solidified PDMS was cut into a 15-by-15 array and used as the master template. The master templates were sterilized in a sterile biosafety cabinet using 70% ethanol with UV light, then washed with cell culture grade water and dried prior to each experiment.

To prepare the multicellular aggregation system, 3% (*w*/*v*) agarose powder (LPS Solution, Daejeon, Republic of Korea) was dissolved by boiling in high-glucose DMEM (Welgene, Korea) in order to provide a non-adhesive substrate. The dissolved agarose-DMEM solution (3 mL) was poured into 35 mm cell-culture dishes (SPL, Pocheon, Republic of Korea) to produce agarose microwell plates. The patterned surface of a sterilized master pattern was carefully placed atop the agarose-DMEM solution to stamp out the microwell array pattern. After solidification, the master pattern was carefully removed and hydrated with cell culture grade water, leaving a 15-by-15 array (225-microwell) patterned agarose-DMEM surface. For cell cultivation, the aggregation system was sterilized in a biosafety cabinet with UV light.

### 2.3. Three-Dimensional Circular Sector Analysis of MAs

For image-based analysis of heterogeneous cell populations, we used CellTracker™ (Thermo Fisher Scientific, Waltham, MA, USA) to distinguish cell types. According to the manufacturer’s manual, CellTracker Green CMFDA Dye, CellTracker Red CMTPX Dye, and CellTracker Blue CMAC Dye were used to stain the U87 and U251 cells and the pdGCs, respectively. After heterotypic cultivation for 24 h, the self-assembled MAs were inspected under a confocal microscope (Nikon, Tokyo, Japan). The 3D spatial distribution of the MAs was quantitatively characterized by dividing a single 500 μm microwell into five concentric zones. The target pixel area in each image was calculated separately for each channel.

### 2.4. Proliferation and Drug-Resistance Assays

We used the conventional WST-1 method and EZ-Cytox assay (water-soluble tetrazolium salt method) to evaluate cell proliferation and viability. Briefly, pdGCs were incubated with 10 μL of EZ-Cytox solution for 3 h. After the formation of formazan crystals, the absorbance at 550 nm of the culture supernatant was measured using a microplate reader (Bio-Rad, Hercules, CA, USA). Cell proliferation and viability were measured based on the optical density compared to non-reacted EZ-Cytox solution.

### 2.5. Characterization of pdGC Invasion

pdGCs were first seeded for 24 h to promote multicellular aggregation. In order to create microscale niches, the pdGCs were embedded in a hyaluronic acid-rich ECM hydrogel to evaluate their invasiveness. Briefly, a 4 mg/mL collagen hydrogel was prepared on ice immediately prior to use by diluting stock collagen type I from rat tails (Corning, NY, USA) with hyaluronic acid solution (Lifecore, Chaska, MN, USA). NaOH (1 M) was added to raise the pH to 7.4. The solutions were mixed thoroughly prior to gelation on pdGCs. Hydrogels were incubated at 37 °C for 1 h prior to the addition of cell culture medium. For every 24 h period, cells invading the ECM hydrogels were inspected via confocal microscopy (Nikon). The *z*-axis scanned images of the invading pdGCs were reconstructed as 3D images and presented as top and cross-sectional views.

### 2.6. Microarray Experiment

Total RNA was extracted from glioblastoma TSs and their matched patient tissues using a Qiagen RNeasy Plus Mini kit, according to the manufacturer’s protocol, and then loaded onto an Illumina HumanHT-12 v4 Expression BeadChip (Illumina, San Diego, CA, USA). After applying a variance-stabilizing transformation, the data were quantile-normalized using the R/Bioconductor lumi package (https://pubmed.ncbi.nlm.nih.gov/18467348, accessed on 10 October 2022).

### 2.7. RNA-Seq Experiment

We extracted total RNA using the NucleoSpin RNA XS kit (Macherey-Nagel) (Fisher scientific, Waltham, MA, USA) from the first-, second-, and third-screened TS14-48 and TS15-88 pdGC subpopulations, and ensured the quality of RNA using Tapestation 4200 (RIN values > 7.5) for all samples. RNA-seq libraries were constructed using an Illumina Truseq Stranded mRNA Library Prep kit (Illumina, San Diego, CA, USA). Briefly, mRNA was purified using oligo-dT beads and fragmented in an enzymatic reaction. After fragmentation, cDNA was generated by reverse transcription. The cDNA libraries were constructed and sequenced in 75-bp paired-end mode on the Illumina Nextseq 550 platform (Illumina, San Diego, CA, USA). External RNA controls consortium and RNA spike-in mixes (Thermo Fisher, 4456740, Waltham, MA, USA) were included for quality assurance purposes.

### 2.8. Bioinformatics Analysis

Visualization and analysis of gene expression was performed using Morpheus software (https://software.broadinstitute.org/morpheus, accessed on 10 October 2022). GSEA was performed using the Molecular Signatures Database (MSigDB) to classify KEGG pathways. Finally, a GO enrichment analysis with results ranked by *p*-value (http://geneontology.org, accessed on 10 October 2022) was performed.

### 2.9. Statistical Analysis

Data are presented as the mean ± standard error of the mean. Levels of significance for comparisons between two independent samples were determined using Student’s *t*-test. Groups were compared using one-way analysis of variance with Tukey’s post hoc test for significant main effects.

## 3. Results

### 3.1. Three-Dimensional (3D) Heterotypic Cultivation of Heterogeneous Cell Populations Based on Self-Assembly

Figure 2a shows an overall schematic illustration of the assessment procedure. To promote self-assembly, a microwell-based array (15-by-15 array composed of 225 microwells of 500-μm diameter and 500-μm depth) was introduced. When individual cells are aggregated into a single cluster within a microwell, each aggregate may exhibit a distinct spatial configuration resulting from self-assembly events. We hypothesize that the distribution of cells in a microwell would rely on cell affinity, resulting in spatial arrangement of the multicellular aggregation.

In order to quantify morphometric information, we categorized the spatial configuration of cells using five concentric zones, of which the most internal was Zone I and the most external was Zone V (Figure 2a). For example, when cells are distributed near Zone I, we presume that the cell subpopulation is of the internal type. By contrast, cells at the periphery of the aggregate, close to Zone V, are of the external type in terms of spatial configuration. Based on the self-assembly pattern of multicellular aggregates, we sought to identify meaningful correlations between morphological information and other tumor hallmarks, including cell proliferation and invasiveness, drug responses, and genotype.

To validate our approach, we tested two glioblastoma cell lines (U87 and U251), which exhibit distinct intercellular affinities (Figure 2b and Appendix A). Under non-adhesive three-dimensional (3D) culture conditions, aggregates of adjacent cells formed a single cluster of distinct shapes. U251 cells (with strong mutual affinity) formed packed clusters, whereas U87 cells clusters (with weak mutual affinity) exhibited relatively dispersed morphology (Figure 2b). This differential intercellular affinity was also observed in a conventional two-dimensional (2D) culture system (Figure 2b). Such differences between cells impacted the self-assembly pattern. U251–U87 heterotypic multicellular aggregates (MAs) became spatially segregated (Figure 2c–e). U251 cells aggregated internally and U87 cells moved to the periphery within 24 h. Almost 90% of U87 cells were resident in Zones II (48%) and III (45%), while most U251 cells were found in Zones I (57%) and II (42%), as shown in the distribution histogram and the circular sectoral graph (Figure 2d). Thus, cells preferentially sought cells of the same type through space and time.

We then explored whether the self-assembly pattern of aggregates reflected the subsequent course of tumor cell invasion in the presence of cell–ECM interactions. Notably, invasion of heterotypic MAs commenced with the outer U87 cells, followed by the inner U251 cells (Figure 2f and Appendix A). U87 cells in the outer zones led the invasion, followed by U251 cells from the inner zones. A similar tendency of invasive potential was shown by homotypic aggregates of each cell. Dissociated U251 aggregates spread throughout the matrix, whereas U87 cells became individually disseminated from the proliferating aggregate core (Appendix A). Therefore, the 3D spatial configuration was reflected in the invasive behavior of each cell type.

Taken together, the data showed that selective affinity between two different cell subpopulations induced a spatially segregated self-assembly process; thus, the spatial configuration within aggregates can be used as a morphometric indicator of cellular heterogeneity.

### 3.2. Heterotypic Cultivation of Patient-Derived Glioblastoma Cells MAs to Assess Intertumoral Heterogeneity

Next, we introduced unknown-phenotype cells (i.e., patient primary cells) with two pre-characterized cell lines as references (Figure 3a). We tested primary glioblastoma cells from eight patients diagnosed with wild-type IDH1 (Table 1). When the patient-derived glioblastoma cells (pdGCs) were co-cultured with U87 and U251 cells, the cells self-assembled into single clusters within 24 h (Appendix A). Because the affinities of U87 and U251 cells are known, as examined in Figure 2, we divided the spatial configurations of the U87–U251–pdGC MA clusters into three types that reflected their intercellular affinities (Figure 3b). If their affinity was the strongest among cells in the MA, the patient’s primary cells would be found in the MA core, surrounded by U251 and U87 cells (in that order). Conversely, pdGCs located near the outside of MAs would exhibit the weakest cell–cell affinity. If the pdGCs were of intermediate affinity, U87 cells would be on the periphery, U251 cells internal, and pdGCs in between. We analyzed the 3D spatial configurations of all eight pdGCs via circular section analysis (Figure 3c). TS14-48 cells were located at the centers of the MAs, whereas TS15-88 cells were found on the outer surfaces. To semi-quantify the spatial configurations, we ranked all pdGCs from 1 (internal, strongest cell–cell affinity) to 8 (external, weakest affinity) and sub-grouped the pdGC MAs into internal (1–4) and external (5–8) types (Figure 3d).

In order to compare the biological characteristics of cells with varying affinities, we performed microarray data analysis (transcriptome profiling). Hierarchical clustering revealed that the internal and external subgroups exhibited different gene expression profiles (Appendix A). When significant (*p* < 0.05) differential gene expression levels of the internal and external subgroups were compared, the results were diametrically opposed. To further evaluate these differences, we performed gene set enrichment analysis (GSEA) (Figure 3e). The Kyoto Encyclopedia of Genes and Genomes (KEGG) cell cycle, DNA replication, and mismatch repair pathways were upregulated in the external versus the internal subgroup. Notably, these pathways frequently vary among patients with high-grade glioblastomas exhibiting therapeutic resistance. The internal types exhibited upregulated expression of KEGG pathways involving cell adhesion molecules (CAMs), antigen processing and presentation, and focal adhesion. Thus, self-assembly pattern-based morphometric analysis using the multicellular aggregation system revealed strong linkages with gene expression.

### 3.3. Phenotypic Correlations of pdGC MAs to Temozolomide Response

To evaluate the clinical implications, we explored the correlations of the self-assembly patterns of pdGC aggregates with the phenotypes of pdGCs, including temozolomide (TMZ) response, cell proliferation, and cell invasion. TMZ response was evaluated based on dose-dependent cell viability with TMZ therapy, and then ranked by gradient color coding (Figure 4a,d). Cell viability monitoring over time permits the ranking of pdGC proliferation (Figure 4b,d). Invasion of pdGC was also analyzed by calculating the invaded area from 0 h to 72 h, and the rank was derived by quantifying the change in invasion area (Figure 4c,d and Appendix A). Intriguingly, pdGCs that self-assembled at the margins of MAs containing U251 and U87 cells demonstrated elevated levels of drug resistance (Figure 4d). Moreover, the rank by spatial configuration (internal vs. external) and TMZ responses (sensitive vs. resistant) resulted in the same two pdGC groups (Figure 4d), and the Kaplan–Meier curve of spatial configuration and TMZ response revealed that patients with more resistant, external-type pdGCs had a lower survival rate (Figure 4e). The survival rate of patients with internal-type pdGCs was substantially higher than that of patients with the external-type pdGCs (Figure 4e). The Kaplan–Meier curve of the two groups, by the rank of cell proliferation and cell invasion (high vs. low), revealed no statistically significant difference in survival (Figure 4f). The aggregate core had more TMZ-sensitive pdGCs than the outermost layer. Self-assembly patterns of pdGC aggregates correlated positively with the extent of TMZ resistance (Figure 4g). When pdGCs were organized from internal type to external type, the TMZ response heatmap resembled that of the self-assembly phenotype (Figure 4h).

Taken together, the results show that morphometric assessment of self-assembled MAs enables the characterization of inter-patient heterogeneity, in terms of both response to TMZ treatment and patient outcome.

### 3.4. Heterotypic Cultivation of pdGC MAs to Assess Intratumoral Heterogeneity

Next, we investigated whether self-assembly pattern-based assessment enables the elucidation of intratumoral heterogeneity. Because the poor prognosis and treatment failure associated with glioblastoma have been attributed to intratumoral heterogeneity and the presence of diverse cell subpopulations within tumors [1], we hypothesized that pdGC intratumoral heterogeneity might be correlated with the drug response and spatial configuration (Figure 5a). Presumably, a cell subpopulation exhibiting strong self-affinity will aggregate more rapidly, compared to other subpopulations, with the culture.

In order to collect subpopulations derived from single pdGCs, we sorted the cells based on the size of cellular aggregates, which hypothetically depends on the aggregation speed; cells with high affinity have faster cellular aggregates. Cell subpopulations exhibiting strong affinity could aggregate more rapidly compared to other subpopulations. Thus, during pdGC cultivation, we filtered the cellular aggregates by time once they were formed. We filtered the pdGC to separate and screen it with large micropores (>70-μm), denoting the first (1st)-screened. We re-cultivated flowthrough and repeated the filtration procedure to obtain the second (2nd)-screened. We used the last flowthrough as the third (3rd)-screening directly after filtration for the 2nd-screened subpopulation (Figure 5b).

We stained the 1st-, 2nd-, and 3rd-screened pdGC subpopulations with CellTracker (red, green, and blue, respectively) to index the internal and external features (Figure 5c), and evaluated the intratumor heterogeneity using TS14-48, TS13-20, TS15-88, and TS16-139 pdGCs. There was no significant difference in the spatial configurations of TS14-48 and TS13-20, as observed. Only TS15-88 and TS16-139 exhibited distinctive assembly patterns, and imaging confirmed that only the external-type pdGC subpopulations segregated (Figure 5d). We then evaluated the drug responses of those subpopulations. Because aggregate formation is dependent on cell density (Appendix A), we seeded pdGC subpopulations at various densities and then added TMZ. Notably, the responsiveness of the third-screened subpopulations of TS14-48 and TS15-88 cells to TMZ, at all tested concentrations, differed between the two populations (Figure 5e and Appendix A).

We further explored whether sequentially collected pdGC subpopulations exhibited distinct gene expression signatures. First-, second-, and third-screened TS14-48 and TS15-88 pdGC subpopulations were subjected to transcriptome (RNA-sequencing (RNA-seq)) analyses. The scatterplots and Pearson/Spearman correlations between subpopulation global gene expression profiles revealed clear intertumoral differences between TS14-48 and TS15-88 (Appendix A). RNA-seq analysis revealed distinctive transcriptional changes in the third-screened TS15-88 subpopulation (TS15-88_3rd) compared to the first- and second-screened populations (TS15-88_1st and TS15-88_2nd) as well as all TS14-88 subpopulations (TS14-88_1st, TS14-88_2nd, and TS14-48_3rd) (Figure 5f). In total, 178 genes were upregulated and 112 were downregulated in the TS15-88_3rd population. The 29 enriched genes in the TS15-88_3rd population included *COL19A1*, *ITGA2B*, *PDGFB*, *COL8A2*, *ITGA8*, *ITGA1*, *COL5A3*, and *WNT3* (Figure 5g). GSEA revealed significant upregulation of extracellular matrix (ECM)-related genes. KEGG pathway analysis identified genes involved in ECM receptor interaction (Figure 5h). Gene Ontology (GO) analysis revealed that TS15-88_3rd cells exhibited marked upregulation of the biological processes of ECM organization, cell–matrix adhesion, and cell–cell adhesion, as well as the canonical Wnt signaling pathway (Figure 5f). Notably, activation of the latter pathway is highly associated with drug resistance in patients with glioblastoma [14]; pathway enrichment, together with changes in secreted ECM components, in TS15-88_3rd cells may contribute to tumor heterogeneity and drug resistance. We found that genes related to ECM secretion, remodeling, and molecular functions; cellular components; and reactomes were enriched in TS15-88_3rd cells, indicating high-level intratumor heterogeneity among drug-resistant TS15-88 cell populations.

These results suggest that self-assembly-based assessment enables rapid, robust phenotypic screening of heterogeneous cell populations within a same tumor specimen; also in addition, correlations with TMZ resistance were apparent.

## 4. Discussion

In this study, a novel approach for characterizing tumor heterogeneity based on the intrinsic characteristics of glioblastoma is proposed. In a heterogeneous tumor population, the spontaneous formation of multicellular aggregates is dependent on the preferred self-assembly of cells with similar intercellular affinities. Our method identifies preferential intercellular affinity between diverse subpopulations of glioblastoma cells, which correlates strongly with the TMZ response. Therefore, our findings imply that the pattern of multicellular self-assembly can serve as an indicator of glioblastoma heterogeneity, thus allowing the prediction of therapy response.

Our approach has various technical advantages; our multicellular aggregation system may offer spatial information based on intercellular affinity within a relatively short culture period (24 h). This method is based on the self-assembly and evaluation of morphometric features of assembled MAs; both experimental and analytic processes are simple to execute. In addition, this rapid evaluation enables an evaluation of context-based assessment with intrinsic characteristics that are unaltered by the phenotypic plasticity of cells. These advantages outweigh the disadvantages associated with marker-based immunohistochemistry and omics-based approaches.

As a heterogeneous subpopulation with specific characteristics develops during the evolution of a tumor, such heterogeneity impedes the development of effective treatments. Based on association studies between phenotypic information and several tumor hallmarks, we discovered a substantial correlation between the spatial arrangement of the cells and TMZ resistance. In this respect, our technology can provide a clinically relevant approach. The cell population with weak intercellular affinity exhibited a high level of drug resistance; this is critical for patient survival. The patient’s cells can be directly cultured in our system, and their heterogeneity can then be evaluated. On the basis of our intuitive assessment, the spatial configuration of the cells can be used to forecast their chemosensitivity, thus facilitating the establishment of an appropriate therapy approach for patients.

The adhesive affinity of cells for specific ligands is spatiotemporal in nature and determined principally by adhesion molecules [21,22,23,24,25]. Although receptor/ligand interaction analysis of major CAMs (including the immunoglobulin superfamily (IgCAMs), cadherins, and integrins) is typically conducted to evaluate cell–cell affinity, cells change their adhesive properties in a context-dependent manner [26,27]. In this study, we provided heterogeneous cell populations to an anchorage-independent environment in order to assess their intercellular affinity capability. In contrast to the situation involving receptor/ligand interactions, cell–cell contacts play a crucial role in determining the morphological characteristics of self-assembled aggregates, resulting in a collection of cohering cells with similar affinity. Our GSEA results suggested that internal types of cell subpopulations show strong cell affinity in terms of upregulated pathways in CAMs and homotypic cell–cell adhesions. In contrast, external types of cell subpopulations exhibit fewer cell–cell contacts, but high ECM receptor interaction, with high ECM remodeling-related pathways. Since ECM offers structural and biochemical support for tumor malignancies to govern proliferation, self-renewal, and differentiation of stem cell-like subpopulation, in addition to chemoresistance [28,29,30], we collectively suggest that tumor subpopulations with low cell affinity may develop an increased ECM remodeling mechanism that would likely contribute to the resistant response to therapy.

Although our proposed system gives new insight into the analysis of tumor heterogeneity, it has certain drawbacks. Due to the restricted number of patients in this investigation, we were only able to rank the phenotypic association between eight patients, dividing them into two distinct groups for relative comparison. Only the link between TMZ reaction and self-assembly pattern was demonstrated in this study, making it difficult to generalize the correlation between self-assembly pattern and therapeutic response. Therefore, it would be useful to apply this evaluation system to assess other chemotherapeutic drugs, such as Carmustine, Bevacizumab, or other prospective drugs, using a large sample number from glioblastoma patients. In addition, it would be necessary to investigate the mechanism behind the assembly process of hMAs with target gene editing.

## 5. Conclusions

This research offers valuable insights on tumor heterogeneity at both the inter- and intra-cellular levels. Intrinsic cell affinity among diverse tumor subpopulations promotes the spontaneous self-assembly of 3D multicellular aggregates, demonstrating a connection between chemosensitivity and prognosis. For future precision medicine and pharmaceutical applications, our assessment approach can be broadly applied to a variety of heterogeneous tumors.

## Figures and Tables

**Figure 1 cancers-14-05910-f001:**
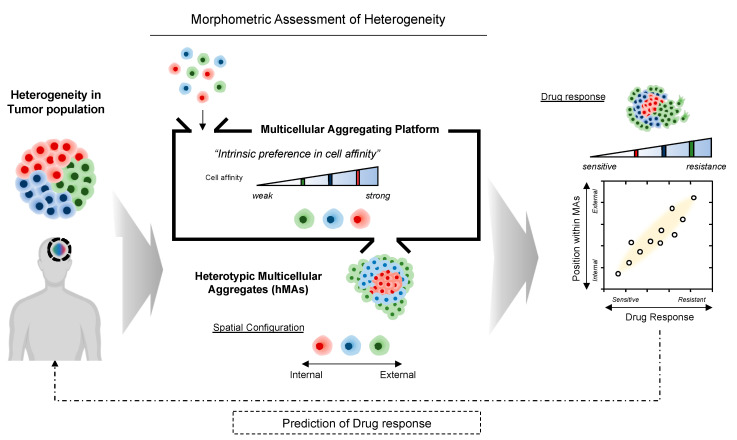
Illustrative overview of morphometric assessment of heterogeneous tumor cell populations based on intercellular affinity preferences. Multicellular aggregation of heterogeneous cell populations may result in a spontaneous 3D spatial configuration of cell aggregates, depending on the differential intercellular affinity.

**Figure 2 cancers-14-05910-f002:**
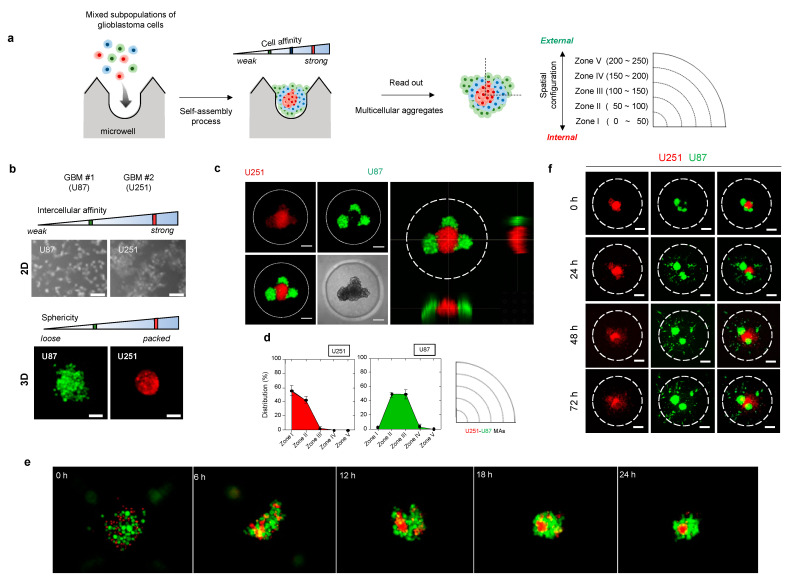
Three-dimensional (3D) heterotypic cultivation for rapid screening of heterogeneous cell populations based on intercellular affinity preferences. (**a**) Schematic overview for assessing the intercellular affinity of patient-derived tumor cells. (**b**) Morphologies of two distinct glioblastoma cell lines in two-dimensional plates and 3D multicellular aggregates. U87MG and U251MG: the two distinct cell lines used in the multicellular aggregation assay system. (**c**) Representative fluorescence images of the 3D spatial distribution of glioblastoma heterotypic multicellular aggregates (MAs). U251 (red), U87 (green). Scale bar: 100 μm. (**d**) Quantitative analysis of fluorescence images of the 3D spatial configurations and the spatial configurations of MAs based on the evaluation of circular zones. (**e**) Time-lapse photographs of the U251–U87 MA assembly. (**f**) Time-lapse of the invasion of U251–U87 heterotypic MAs over 72 h. Scale bar: 100 μm.

**Figure 3 cancers-14-05910-f003:**
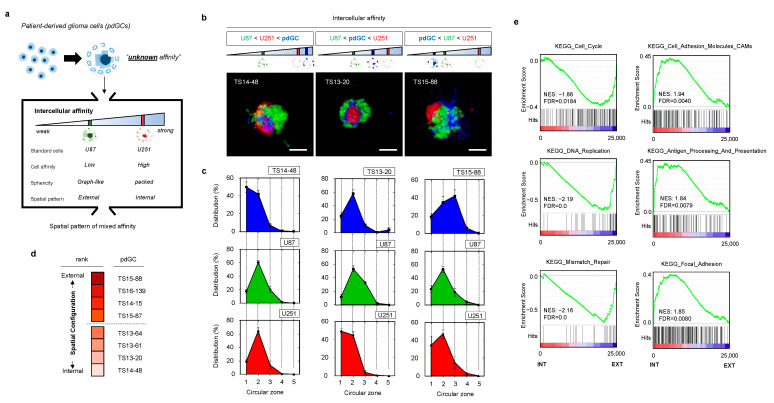
Formation and analysis of MAs of patient-derived glioblastoma cells (pdGCs) and the phenotypic associations thereof. (**a**) Multicellular self-assembly of pdGCs with unknown intercellular affinity compared to those of the U251 and U87 glioblastoma cell lines. (**b**) Representative fluorescence images of pdGC–U251–U87 MAs. Scale bar: 100 μm. (**c**) Spatial configurations (3D) within pdGC–U251–U87 MAs. (**d**) pdGC scoring based on 3D spatial configuration. (**e**). Enrichment score plots of the distinct KEGG phenotypes of the internal and external subgroups.

**Figure 4 cancers-14-05910-f004:**
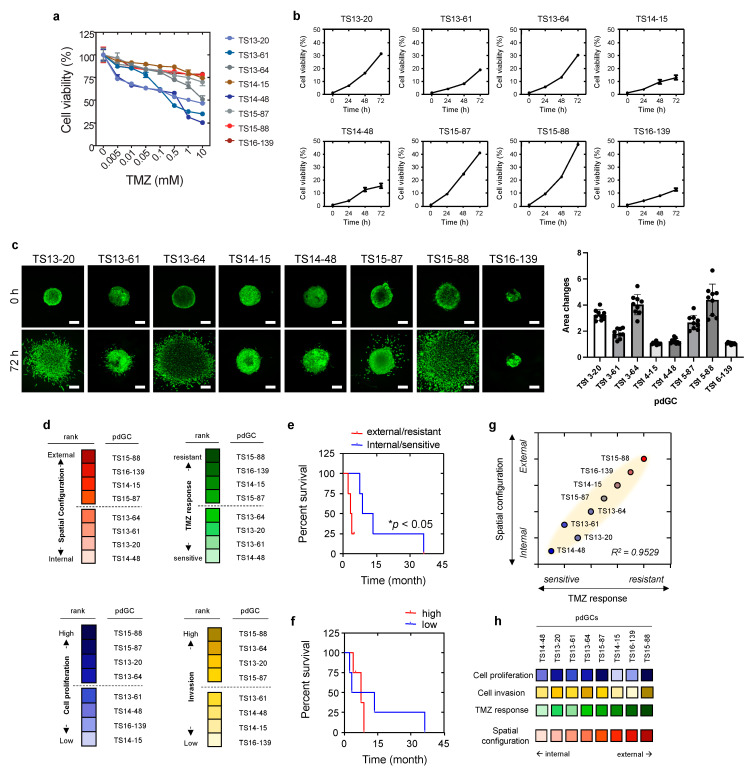
Phenotypic correlations with spatial configuration of pdGCs MAs. (**a**) pdGC viability in response to temozolomide (TMZ). (**b**) Graphs for all pdGCs proliferation over time for 72 h. (**c**) Fluorescent images and quantification of pdGCs after 72-h invasion. Scale bar: 100 μm. (**d**) Ranking of pdGC TMZ response (sensitive to resistance), pdGC proliferation (low to high), and pdGC invasion (low to high) by color-code. (**e**) Kaplan–Meier survival plot of patients whose pdGCs were classified in terms of MA TMZ response and 3D spatial configuration. Statistical significance: * *p* < 0.05. (**f**) The Kaplan–Meier survival graph of patients whose pdGCs were classified in terms of MA proliferation and invasion. (**g**) Correlation between the drug response and 3D spatial configuration. (**h**) Comparative pdGC heatmap of the associations of cellular proliferation and invasion, as well as TMZ response, with 3D spatial configuration in MAs.

**Figure 5 cancers-14-05910-f005:**
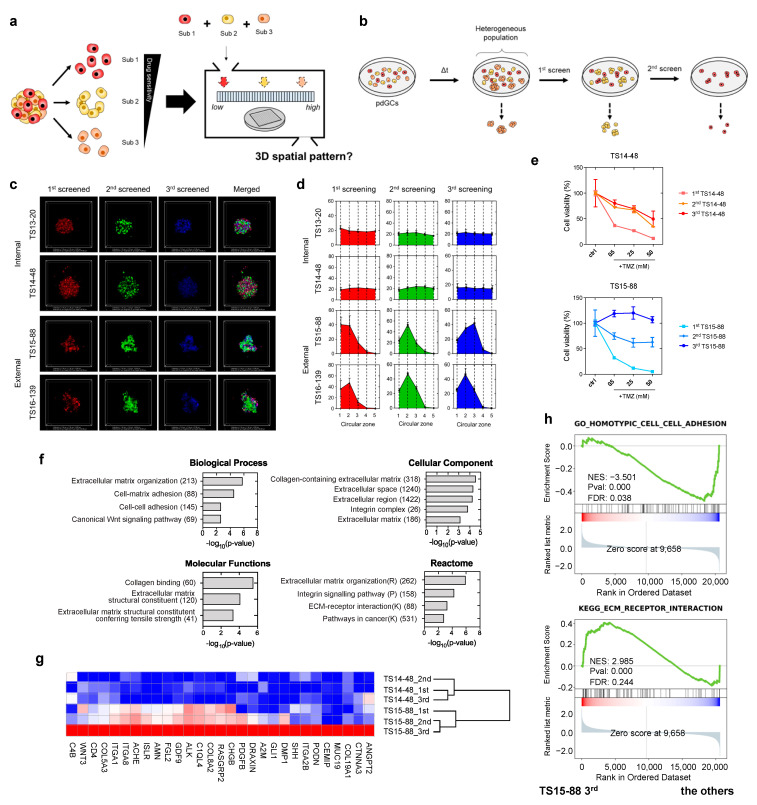
Multicellular self-assembly assay of intratumoral heterogeneity in drug response. (**a**) Schematic of the screening process used to identify phenotypic subpopulations within a single pdGC population. (**b**) Screening of three subpopulations from a single pdGC population. (**c**) Representative fluorescence images showing the formation of 3D spatial configurations in pdGC-only MAs. Scale bar: 200 μm. (**d**) Spatial configurations (3D) of pdGC–U251–U87 MAs. (**e**) TS14-48 and TS15-88 cell viabilities (TMZ resistance levels); the subpopulations were those of single pdGCs (initial density 1.0 × 10^5^ cells). (**f**) A bar graph showing the numbers of upregulated genes in TS15-88_3rd cells, compared to the other subpopulations (TS14-48 1st, 2nd, 3rd and TS15-88 1st, 2nd), employing Gene Ontology terms in terms of biological process, cellular components, molecular functions, and reactome. (**g**) Enriched gene expression heatmaps of sequentially screened TS14-88 and TS15-88 pdGCs. (**h**) Gene set enrichment analysis score plots for distinct phenotypes in the third-screened TS15-88 subpopulation (TS15-88_3rd), compared to the other subpopulations.

**Table 1 cancers-14-05910-t001:** List of patient-derived glioblastoma cells (pdGCs) used in this study.

pdGC	Pathology	Age	Gender	IDH1 Mutation	MGMT Promoter
TS13-20	Glioblastoma	61	M	Wild type	Methylated
TS13-61		57	M		Unmethylated
TS13-64		56	F		Unmethylated
TS14-15		67	M		Methylated
TS14-48		42	F		Unmethylated
TS15-87		75	F		Unmethylated
TS15-88		61	M		Unmethylated
TS16-139		62	M		Unmethylated

## Data Availability

The raw microarray data were deposited in the Gene Expression Omnibus database under accession number GSE159000. RNA-seq sequence data that support the findings of this study have been deposited into the NCBI Sequence Read Archive under the BioProject ID PRJNA691140. All other data are available from the corresponding authors or first author upon reasonable request.

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
