# Peer review of "Assessing Spatial Distribution of Multicellular Self-Assembly Enables the Prediction of Phenotypic Heterogeneity in Glioblastoma"

_cancers, 2022, doi:10.3390/cancers14235910_

Round 1

Reviewer 1 Report

In their manuscript "Assessing Spatial Distribution of Multicellular Self-Assembly Enables to Predict the Phenotypic Heterogeneity in Glioblastoma" Cha and co-workers describe a method to reveal the cellular heterogeneity of human glioblastomas in cell culture. This is the description of an interesting approach. Therefore, it is worth publishing with Cancers in the section Methods/Technology.

There are, however, few questions, that should be addressed by the authors in revision.

1. Since it is a technical paper for the Methods section it should describe all methods that are used. In Mat/Met the authors refer several times to previous work (e.g. first sentence 2.1. and 2.2.). The method should be completely described in the paper.

2. The authors try to focus on the biological consequences of the results of their experiments. Therefore, they investigated the transcriptomes of the pdGCs from different regions. But it is not clear if the differences are the cause or the consequence of the different distribution. The authors should knock-out or overexpress important genes to show the consequences for the spatial distribution. 

3. The TMZ experiments are of clinical relevance and therefore very interesting. Once again, it would be highly interesting, if the identified genes are responsible for TMZ resistance and/or spatial distribution. Therefore, knock-out/overexpression experiments would be highly interesting.

4. The discussion is very superficial, not even more than an abstract. The authors should discuss the technical aspects of the method, should discuss the biological results, describe some outlook of possible fields of experiments and should discuss alternative methods.

5. There are several typing errors etc. including mistyping of chapters (e.g. 3.23 D heterotypic ...)

Author Response

  1. Since it is a technical paper for the Methods section it should describe all methods that are used. In Mat/Met the authors refer several times to previous work (e.g. first sentence 2.1. and 2.2.). The method should be completely described in the paper.

Our response: We elaborated the details about the methods we used for this study in the revised version of the manuscript.

  1. The authors try to focus on the biological consequences of the results of their experiments. Therefore, they investigated the transcriptomes of the pdGCs from different regions. But it is not clear if the differences are the cause or the consequence of the different distribution. The authors should knock-out or overexpress important genes to show the consequences for the spatial distribution.

Our response: Our study is suggesting a new approach aiming the characterization of the tumor heterogeneity based on the inherent traits of the pdGCs. Since loss of the intercellular affinity is highly associated with the phenotypic changes, enabling the cells to detach and invade, we assumed that there are different intercellular affinities in heterogenous tumor population, spontaneously resulting in multicellular sub-aggregates due to preferential self-assembly between the cells with similar affinities. Our multicellular aggregation system was able to provide spatial information based on intercellular affinity within the relatively short-term culture period (~24 hr). This fast assessment enables to characterize tumor heterogeneity, as well as to overcome the drawbacks, such as phenotypic plasticity of patient-derived cells. We utilized the transcriptomes of the pdGCs to analyze the common characteristics of pdGCs, ranked by spatial distribution in our system. We can demonstrate that pdGCs located at the outermost regions (external) of heterotypic multicellular aggregates (hMAs) shows distinct transcriptomic expressions from the pdGCs located innermost regions (internal) of hMAs. From our gene set enrichment analysis (GSEA), the external pdGCs showed upregulated pathways including KEGG cell cycle, DNA replication, and mismatch repair pathways, while the internal pdGCs exhibited cell adhesion molecules (CAMs), antigen processing and presentation, and focal adhesion. Thus, we suggest that self-assembly pattern-based morphometric analysis using the multicellular aggregation system revealed strong linkages with gene expression. As suggested, knock-out/overexpression of candidate genes would be of great interest to investigate the detail mechanism in the assembly process of hMAs. We here focus on explaining our new approach, how to fast-screen the tumor heterogeneity using the multicellular aggregation system, so it would be valuable to explore underlying mechanism with target gene editing as a next step. We also stated this to the discussion part in the revised version of our manuscript for further understanding and suggestion.

  1. The TMZ experiments are of clinical relevance and therefore very interesting. Once again, it would be highly interesting, if the identified genes are responsible for TMZ resistance and/or spatial distribution. Therefore, knock-out/overexpression experiments would be highly interesting.

Our response: Again, we appreciate the valuable suggestion. We explored the correlations of the self-assembly patterns of pdGC aggregates with various pdGC phenotypes including temozolomide (TMZ) response, cell proliferation and cell invasion, to evaluate the clinical implications. Interestingly, the self-assembly patterns of pdGC aggregates correlated positively with the extent of TMZ resistance, but not with the invasive or proliferation capacity. pdGCs in the aggregate core were more TMZ-sensitive than those in the outermost layer. The Kaplan–Meier curve of TMZ response showed that patients exhibiting higher resistance experienced poorer survival. Moreover, patients with internal-type pdGCs survived for significantly longer than those with external-type pdGCs. Therefore, our results suggest that multicellular self-assembly pattern can be an indicative of glioblastoma heterogeneity, further predicting the therapeutic response. In addition to our response to point 2, we here focus on explaining our new approach, how to fast-screen the tumor heterogeneity using the multicellular aggregation system, so it would be valuable to explore underlying mechanism with target gene editing as a next step.

  1. The discussion is very superficial, not even more than an abstract. The authors should discuss the technical aspects of the method, should discuss the biological results, describe some outlook of possible fields of experiments and should discuss alternative methods.

Our response: Based on the valuable comments, we describe more explanation on the technical aspects of the method, discuss the biological results, outlook of possible fields of expression, possible next steps, and limitations of the study in the revised version of our manuscript.

  1. There are several typing errors etc. including mistyping of chapters (e.g. 3.23 D heterotypic ...)

Our response: As the reviewer commented, we carefully corrected the typing errors in the revised version of the manuscript.

Reviewer 2 Report

The manuscript “Assessing Spatial Distribution of Multicellular Self-Assembly Enables to Predict the Phenotypic Heterogeneity in Glioblastoma” represents a truly innovative original study on glioblastoma, known for its dismal prognosis even in patients receiving the best possible treatment. Authors have developed an original self-assembly-based cellular system to assess tumour heterogeneity and the cardinal features of neoplasia, including proliferation, invasiveness, response to treatment and gene expression profile. The ultimate clinical value of such systems could be in the capability to predict the therapeutic response. The scientific significance is equally high, including deeper understanding of pathogenesis of heterogeneity and development of a new tool to assess it.

The applied methods are innovative and comprehensively described. The results are explicitly presented and supplemented by outstanding figures. The manuscript is characterised by high level of English language, ensuring good scientific comprehensibility; it is well-written and interesting.

I have only few remarks:

Although the article is outstanding from the scientific point of view, some formatting issues should be solved, e.g., formatting of References, formatting of reference numbers in the text and formatting of the contacts for the corresponding author.

In the Introduction, authors have noted that there are no assessment technologies that can predict intratumoural heterogeneity within the same tumour specimen (lines 60 – 61). Actually, the heterogeneity in protein level can be assessed via immunohistochemistry, and this method has been used for molecular classification of glioblastoma. See, please, “Molecular classification of diffuse gliomas” by Jakovlevs A et al., Pol J Pathol, 2019;70(4):246-258. PMID: 32146793; and outline shortly the benefits of your method over immunohistochemical assessment.

Finally, I would like to thank the authors for their huge work input. It was a true honour and a great pleasure to review this manuscript.

Author Response

  1. Although the article is outstanding from the scientific point of view, some formatting issues should be solved, e.g., formatting of References, formatting of reference numbers in the text and formatting of the contacts for the corresponding author.

Our response: Thank you for the detailed comments. As the reviewer mentioned, we carefully inspect the manuscript and corrected the formats in the revised version of the manuscript.

  1. In the Introduction, authors have noted that there are no assessment technologies that can predict intratumoural heterogeneity within the same tumour specimen (lines 60 – 61). Actually, the heterogeneity in protein level can be assessed via immunohistochemistry, and this method has been used for molecular classification of glioblastoma. See, please, “Molecular classification of diffuse gliomas” by Jakovlevs A et al., Pol J Pathol, 2019;70(4):246-258. PMID: 32146793; and outline shortly the benefits of your method over immunohistochemical assessment.

Our response: We revised the sentences that explain the benefits of our methods over other methods to avoid the overstatement. As commented, immunohistochemistry has been used as a broadly applicable technique to detect diverse subtypes by the molecular characteristics [1,2]. Immunostaining of specific markers enables to classify the subtypes in glioblastoma, including “classical”, “mesenchymal” and “proneural”. However, immunohistochemistry is based on the selective markers to define each subtype, so it relies on how to select the protein markers for each subtype. Also, it is site-dependent, so the marker expression can be highly influenced by the process of sample preparation. In contrast, our method can overcome these drawbacks, enabling context-based functional assessment with intrinsic features, not depending on the selective markers. Finally, we also would like to thank the reviewer for giving us invaluable comment for a better manuscript.

References

  1. Liesche-Starnecker, F.; Mayer, K.; Kofler, F.; Baur, S.; Schmidt-Graf, F.; Kempter, J.; Prokop, G.; Pfarr, N.; Wei, W.; Gempt, J.; Combs, S.E.; Zimmer, C.; Meyer, B.; Wiestler, B.; Schlegel, J. Immunohistochemically Characterized Intratumoral Heterogene-ity Is a Prognostic Marker in Human Glioblastoma. Cancers 2020, 12, 2964. https://doi.org/10.3390/cancers12102964
  2. Bergmann, N.; Delbridge, C.; Gempt, J.; Feuchtinger, A.; Walch, A.; Schirmer, L.; Bunk, W.; Aschenbrenner, T.; Liesche-Starnecker, F.; Schlegel, J, The Intratumoral Heterogeneity Reflects the Intertumoral Subtypes of Glioblastoma Mul-tiforme: A Regional Immunohistochemistry Analysis. Front Oncol. 2020, 24;10:494. https://doi.org/10.3389/fonc.2020.00494

Reviewer 3 Report

Cha et al present an interesting approach based on the natural segration pattern of cell population. They correlate this segregation with the tumor heterogeneity, response to temozolomide and patient prognosis. Although this study has the merit of a original approach, in my opinion, there are some flaws in the authors' assumption that may impact the research's conclusions. Specifically, it is assumed that cells that segregate together are more alike phenotypically and/or at the transcriptomic level. This is not necessarily true, certain cell types might depend on others to support their growth and proliferation. It is also assumed that cells that segregate inside spheroids are more homogeneous and alike. Again, this might also be influenced by cell size, anchorage-independant growth capacity, differenciation level etc. Therefore, I believe that more studies and analysis are necessary to support the authors' conclusions, and that certain assumptions might need to be removed from the article to maintain its scientific soundness.

Specific comments:

Simple summary: the sentence "peripheral hMA cells (...)" is not clear, please rephrase

Introduction:  the sentence "differential intercellular affinity can be a marker of intratumoral heterogeneity". This is an assumption that needs to be clearly demonstrated in the study.

Results:

Line 169: "in principle, lower-affinity cells spontaneously move to the periphery and higher-affinity cells to the core" From a modelistic point of view, yes, but affinity does not necessarily mean homogeneity (see my comment above).

Figure 2c: if one was to follow the circumference of the spheroid formed by U87/U251, some of that circumference is made by U251 cells also even if they are located closer to the center of the aggregate. This should be taken into account, not only the distance from the aggregate center.

Figure 2f: Was this quantified? Please provide quantifications and stats

Line 184: the word "predict" is not right, probably better to use "presume" or "determined"

Line 194: graph-like, I don't understand this expression. Grape-like?

RNAseq line 243-253: could the authors assess population heterogeneity using the RNAseq data, to see of any correlation is observed with the spatial configuration score?

Line 253: internal cell types have increased CAM, focal adhesion and cell-cell/stroma pathways, generally linked to cells that are more differentiated. This could suggest that internal cells are more differentiated, and could explain their greater sensitivity to temozolomide (fig 4d).

Figure 4c: place this panel after f-k to maintain a logical order. Perform correlation statistics. Change color code so that spatial configuration keeps its own color codes and the other parameters have different codes. In fact, the color code is not shown in 4c. Finally organize the cell lines from internal to external to have a more visual representation

Figure 4d and e are the same since the groups are the same. Is is relevant to show both? Same for 4 i and k

Figure 4g: please quantify and provide stats.

Line 291: I don't understand how the subpopulations were collected using screens. Please rephrase

Figure 5d: it is in fact only the 3rd screen of TS15-88 that is different from the rest. The 3 screens from TS16-139 segregate similarly, all in zone 2, which is a bit strange (how can all 3 populations be more present in this zone?).  Could the authors analyse self-clustering (ie tendancy to cluster with each other) rather than the position?

Figure 5e: the data from the 3 screens from one cell line should be shown here rather than a comparison of each screen for both cell lines. This will show that screen 3 from TS15-88 becomes resistant to tmz, and that the 3 screens of TS16-139 maintain the same level of response to tmz.

Figure legend 5 f et g: indicate what is the 3rd screen compared to in this analysis

Line 331: cells from the 3rd screen (external type) express pathways linked to ecm organization, cell-cell/matrix adhesion which are typical of more differentiated cells. This is in apparent contradiction to data from figure 3e where cells from the internal types express these pathways.

Discussion: please detail how the described methodology would be applied in clinical practice, and what information would it bring to inform the patient's treatment

Author Response

Specific comments:

Simple summary: the sentence "peripheral hMA cells (...)" is not clear, please rephrase

Our response: We revised the sentence “peripheral hMA cells ~” into “The cells located at the outermost hMA ~” for clear expression.

Introduction:  the sentence "differential intercellular affinity can be a marker of intratumoral heterogeneity". This is an assumption that needs to be clearly demonstrated in the study.

Our response: As commented, we added “we assumed that ~” in front of the sentence to indicate that this is a key assumption we would like to demonstrate in the study.

Results:

Line 169: "in principle, lower-affinity cells spontaneously move to the periphery and higher-affinity cells to the core" From a modelistic point of view, yes, but affinity does not necessarily mean homogeneity (see my comment above).

Our response: As commented, to avoid the overstatement, we revised the sentences that describe our core assumption for our system. Specifically, we assumed that the cells may be located in a microwell depending on the cell affinity, resulting in spatial configuration of the multicellular aggregate.

Figure 2c: if one was to follow the circumference of the spheroid formed by U87/U251, some of that circumference is made by U251 cells also even if they are located closer to the center of the aggregate. This should be taken into account, not only the distance from the aggregate center.

Our response: Figure 2c displays the representative image of 3D spatial distribution of glioblastoma heterotypic multicellular aggregates (MAs) by U251 (red) and U87 (green). This provides a clear segregated pattern of both U251 MA and U87 MA. Initially, we indicated the boundary between U251 MA and U87 MA by following the circumference based on the fluorescent signal of U251 MA, but we removed it to just display how the U251-U87 MAs look like in the revised version of the manuscript.

Figure 2f: Was this quantified? Please provide quantifications and stats

Our response: We attached quantification graph for U251-U87 Mas invasion in our supporting information as Fig. S5.

Line 184: the word "predict" is not right, probably better to use "presume" or "determined"

Our response: We revised the expression “predict” to “presume”.

Line 194: graph-like, I don't understand this expression. Grape-like?

Our response: We deleted the expression “graph-like” for clear expression and revised the sentence: U251 cells (with strong mutual affinity) formed packed clusters, whereas U87 cells clusters (with weak mutual affinity) exhibited relatively dispersed morphology

RNAseq line 243-253: could the authors assess population heterogeneity using the RNAseq data, to see of any correlation is observed with the spatial configuration score?

Our response: In this study, we only used microarray data to characterize the genomic expression profiles of pdGCs but did not try RNA sequencing of pdGCs. Instead, we later analyzed subpopulations of the representative internal type pdGC (here, TS14-48) and external type pdGC (here, TS15-88) for transcriptomic analysis of intra-tumor heterogeneity. Assessing the heterogeneity based on RNAseq results could be of interest to find if there is a correlation with the spatial configuration as a further study.

Line 253: internal cell types have increased CAM, focal adhesion and cell-cell/stroma pathways, generally linked to cells that are more differentiated. This could suggest that internal cells are more differentiated, and could explain their greater sensitivity to temozolomide (fig 4d).

Our response: We appreciate the critical comment. We added this explanation in the discussion part of the revised manuscript.

Figure 4c: place this panel after f-k to maintain a logical order. Perform correlation statistics. Change color code so that spatial configuration keeps its own color codes and the other parameters have different codes. In fact, the color code is not shown in 4c. Finally organize the cell lines from internal to external to have a more visual representation

Figure 4d and e are the same since the groups are the same. Is is relevant to show both? Same for 4 i and k

Figure 4g: please quantify and provide stats.

Our response: Thank you for the careful comment about Figure 4. Please find the following response to each comment.

  • To provide a more visual representation, we reorganize the figure panels to maintain the logical order. We first placed the phenotypic characterization such as TMZ response, cell proliferation and cell invasion, and then displayed the rank by each phenotype.
  • For the correlation graph, we added the correlation coefficient (R2=0.9525).
  • We changed the color code for each phenotype, so as to clearly represent each parameter.
  • For the comparative pdGC heatmap, we arrange the pdGCs based on the spatial configuration (from the innermost to outermost), so it enables to easily compare the color-coded heatmap of each phenotype.

Line 291: I don't understand how the subpopulations were collected using screens. Please rephrase

Our response: We supplemented the explanation about the process of collecting subpopulation in the revised version of the manuscript. Here is the process to collect the subpopulation from single pdGC: we filtered the pdGC to separate and screen the bigger sized pdGC subpopulation (>70-μm micropores), denoting first (1st)-screened. We again cultured flowthrough and repeat the filtration process to collect the second (2nd)-screened. We used last flowthrough as third (3rd)-screened.

Figure 5d: it is in fact only the 3rd screen of TS15-88 that is different from the rest. The 3 screens from TS16-139 segregate similarly, all in zone 2, which is a bit strange (how can all 3 populations be more present in this zone?).  Could the authors analyse self-clustering (ie tendancy to cluster with each other) rather than the position?

Our response: The purpose of this analysis is to demonstrate whether the cell subpopulations from single pdGC create hMAs based on self-assembly. Here, Figure 5d shows the distribution of each subpopulation. From the analysis, we can demonstrate that subpopulations from two external types of pdGCs, TS15-88 and TS16-139 exhibit the self-assembled pattern, while subpopulations from two internal types of pdGCs (TS13-20 and TS14-48) displayed no distinct pattern of self-assembly. Certainly, subpopulations from TS16-139 didn’t show clear self-assembly pattern as much as the subpopulation of TS15-88, as the cells are located more loosely throughout the zone 2. That is why the distribution of subpopulation are located in mostly in zone 2, not like TS15-88 subpopulation. Nevertheless, they showed self-assembly pattern, comparing to the subpopulation from internal types of pdGCs. Since the clearest self-assembly pattern within intratumor population was observed in TS15-88 pdGCs, we performed TMZ response test using subpopulation of TS15-88. We believe that this demonstration should be sufficient to intratumor self-assembly tendencies in external types of pdGCs.

Figure 5e: the data from the 3 screens from one cell line should be shown here rather than a comparison of each screen for both cell lines. This will show that screen 3 from TS15-88 becomes resistant to tmz, and that the 3 screens of TS16-139 maintain the same level of response to tmz.

Our response: As suggested, we revised the graph in Figure 5e to demonstrate the clear comparison between 3rdscreened TS15-88 and 3rd screened TS14-48.

Figure legend 5 f et g: indicate what is the 3rd screen compared to in this analysis

Our response: We specify the comparison group (TS14-48 1st, 2nd, 3rd, and TS15-88 1st, 2nd) of TS15-88 3rd for this analysis in the figure legend 5.

Line 331: cells from the 3rd screen (external type) express pathways linked to ecm organization, cell-cell/matrix adhesion which are typical of more differentiated cells. This is in apparent contradiction to data from figure 3e where cells from the internal types express these pathways.

Our response: For investigating the intratumor heterogeneity, we collected the subpopulation from single pdGC by filtering the pdGC to separate and screen the bigger sized pdGC subpopulation (>70-μm micropores), that presumably possesses high cell affinity within the subpopulation. We repeat the filtration process to collect the second (2nd)-screened and the third (3rd)-screened, so we aimed to compare different subpopulation within a single tumor line based on cell affinity. As presented in Fig. 5f, we would like to suggest what difference is found from TS15-88_3rd cell population, which showed significant changes in TMZ response, comparing to the rest of subpopulation groups (TS15-88 1st, 2nd, TS14-48 1st, 2nd, and 3rd). Indeed, Gene Ontology (GO) analysis revealed that TS15-88_3rd cells exhibited marked upregulation of ECM remodeling-related biological process, as well as the canonical Wnt signaling pathway, comparing to the rest of subpopulation groups. In addition, ECM-related cellular components such as collagen-containing extracellular matrix and integrin complex (its binding proteins) are significantly upregulated in TS15-88_3rdsubpopulation. In addition, as described in Fig. 5h, the gene set enrichment analysis (GSEA) also showed upregulated KEGG_ECM receptor interaction (NES=2.985, FDR=0.244), while resulted in downregulated trend in KEGG_cell adhesion molecules (CAMs) (NES=-2.159, FDR=0.587). With the GSEA result of KEGG_cell adhesion molecules (CAMs), we initially intended to show the similar trends of the rest of subpopulation groups with internal types of pdGCs as we analyzed in our intertumor heterogeneity (shown in Fig. 3e). However, when considering our FDR threshold = 0.25, this trend is less significant. So, it would be better to remove this GSEA panel here, and display KEGG_ECM receptor interaction only to show the consistency with GO analysis.

Discussion: please detail how the described methodology would be applied in clinical practice, and what information would it bring to inform the patient's treatment

Our response: As commented, we detailed more discussion on how our method would be applied in clinical practice, and what information would it bring to inform the patient’s treatment in the revised version of our manuscript.

Round 2

Reviewer 1 Report

The corrections and additions made by the authors are appropriate. 

Author Response

Comments and Suggestions for Authors:

The corrections and additions made by the authors are appropriate.

Our response: We would like to thank the reviewer for giving us invaluable input for a better manuscript.

Reviewer 3 Report

The authors did not adress my main concern regarding this study:

"Specifically, it is assumed that cells that segregate together are more alike phenotypically and/or at the transcriptomic level. This is not necessarily true, certain cell types might depend on others to support their growth and proliferation. It is also assumed that cells that segregate inside spheroids are more homogeneous and alike. Again, this might also be influenced by cell size, anchorage-independant growth capacity, differenciation level etc."

**Please demonstrate that cells that are segregated together are more homogeneous at the transcriptomic or phenotypic level than cells that do not segregate together

Regarding my comment:

"Line 291: I don't understand how the subpopulations were collected using screens. Please rephrase"

 And the author's response: We supplemented the explanation about the process of collecting subpopulation in the revised version of the manuscript. Here is the process to collect the subpopulation from single pdGC: we filtered the pdGC to separate and screen the bigger sized pdGC subpopulation (>70-μm micropores), denoting first (1st)-screened. We again cultured flowthrough and repeat the filtration process to collect the second (2nd)-screened. We used last flowthrough as third (3rd)-screened

**Does this mean that cells are sorted based on size?

**How long between the 3 screens?

Regarding my comment:

"Line 331: cells from the 3rd screen (external type) express pathways linked to ecm organization, cell-cell/matrix adhesion which are typical of more differentiated cells. This is in apparent contradiction to data from figure 3e where cells from the internal types express these pathways."

And the author's response: For investigating the intratumor heterogeneity, we collected the subpopulation from single pdGC by filtering the pdGC to separate and screen the bigger sized pdGC subpopulation (>70-μm micropores), that presumably possesses high cell affinity within the subpopulation. We repeat the filtration process to collect the second (2nd)-screened and the third (3rd)-screened, so we aimed to compare different subpopulation within a single tumor line based on cell affinity. As presented in Fig. 5f, we would like to suggest what difference is found from TS15-88_3rd cell population, which showed significant changes in TMZ response, comparing to the rest of subpopulation groups (TS15-88 1st, 2nd, TS14-48 1st, 2nd, and 3rd). Indeed, Gene Ontology (GO) analysis revealed that TS15-88_3rd cells exhibited marked upregulation of ECM remodeling-related biological process, as well as the canonical Wnt signaling pathway, comparing to the rest of subpopulation groups. In addition, ECM-related cellular components such as collagen-containing extracellular matrix and integrin complex (its binding proteins) are significantly upregulated in TS15-88_3rdsubpopulation. In addition, as described in Fig. 5h, the gene set enrichment analysis (GSEA) also showed upregulated KEGG_ECM receptor interaction (NES=2.985, FDR=0.244), while resulted in downregulated trend in KEGG_cell adhesion molecules (CAMs) (NES=-2.159, FDR=0.587). With the GSEA result of KEGG_cell adhesion molecules (CAMs), we initially intended to show the similar trends of the rest of subpopulation groups with internal types of pdGCs as we analyzed in our intertumor heterogeneity (shown in Fig. 3e). However, when considering our FDR threshold = 0.25, this trend is less significant. So, it would be better to remove this GSEA panel here, and display KEGG_ECM receptor interaction only to show the consistency with GO analysis.

**The authors did not adress the apparent contradiction that I pointed out.

The other comments were adressed adequately.

Author Response

Comments and Suggestions for Authors:

The authors did not adress my main concern regarding this study:

"Specifically, it is assumed that cells that segregate together are more alike phenotypically and/or at the transcriptomic level. This is not necessarily true, certain cell types might depend on others to support their growth and proliferation.

It is also assumed that cells that segregate inside spheroids are more homogeneous and alike. Again, this might also be influenced by cell size, anchorage-independant growth capacity, differenciation level etc."

**Please demonstrate that cells that are segregated together are more homogeneous at the transcriptomic or phenotypic level than cells that do not segregate together

Our response: Thank you for the critical comment. Our response: Cell affinity has been suggested as a concept referring to the propensity to interact preferentially with cells of the same type [1]. Depending on the cell affinity, the cells in a mixed aggregate progressively sort to form a well-segregated population by recognizing the identity of their neighbors, gathering with cells of the same type [2]. The cell-cell adhesion molecules (CAMs) that bind specifically to themselves (homotypic binding) have been a molecular explanation for cell-cell affinity and tissue segregation [3]. For instance, multicellular spheroid formed a self-assembled structure due to different gene expression levels of cell affinity, especially E-cadherin. [4] Specifically, this study showed that higher expression of E-cadherin leads to their segregation into the insider layer of the self-organized spheroid, forming a structure with distinct spatial compartments. Thus, the reference suggests cell affinity can determine the spatial configuration during the self-assembly process for the cell spheroid. Likewise, we assumed that different cell affinities in a heterogenous tumor population might induce self-assembled aggregates. Culturing two different glioblastoma cells (relatively, U251MG with higher cell affinity and U87MG with lower cell affinity, described in Fig. S1, S3) showed clear segregation by sorting and gathering themselves and consequently creating spatial configuration in multicellular aggregate: U251MG inside and U87MG outside. Our result suggests that cells with similar affinity can preferentially form aggregates. Likewise, we could determine the location of pdGC in multicellular aggregate by its affinity, compared to the location of the standard cell line, U251MG, and U87MG. From the transcriptomic analysis of the pdGCs, we analyzed the common characteristics of pdGCs. pdGCs at the outermost of the aggregates (hMAs) exhibited distinct transcriptomic profiles from the pdGCs at the innermost regions of hMAs. Based on our gene set enrichment analysis (GSEA), the internal pdGCs exhibited upregulated KEGG pathways in cell adhesion molecules (CAMs). In contrast, the external pdGCs showed no significant upregulations in related pathways (Fig. 3e). Specifically, from our microarray data, the CDH3 gene (cadherin-3, or P-cadherin), a member of cell adhesion molecules, is highly expressed in the internal group, while significantly less in external (Fig. R1). Therefore, we suggest that the self-assembled pattern by the cells has strong linkages with gene expression.

There could be a possibility to have parameters that control the process of self-assembly, including heterotypic interaction (contacting between cells of the different types), cellular plasticity, cell size, anchorage-independent growth capacity, and differentiation level, as commented. This system could provide spatial information based on intercellular affinity within the relatively short-term culture period (~24 hr). Therefore, the self-assembly depends on the intrinsic propensity to interact with cells of similar affinity, leading to segregation in a mixed population. For a better understanding, knock-out/overexpression of candidate genes (possibly, CDH3 gene) would be of great interest to investigate the detailed mechanism in the assembly process of hMAs. To introduce the concept and idea of our study, we added the background details about cell affinity in the introduction part of our revised manuscript.

Fig. R1. Relative expression level of CDH3 gene in internal group and external group.

Regarding my comment:

"Line 291: I don't understand how the subpopulations were collected using screens. Please rephrase"

 And the author's response: We supplemented the explanation about the process of collecting subpopulation in the revised version of the manuscript. Here is the process to collect the subpopulation from single pdGC: we filtered the pdGC to separate and screen the bigger sized pdGC subpopulation (>70-μm micropores), denoting first (1st)-screened. We again cultured flowthrough and repeat the filtration process to collect the second (2nd)-screened. We used last flowthrough as third (3rd)-screened

**Does this mean that cells are sorted based on size?

Our response: Yes. The sorting process is based on the size of cellular aggregates, which hypothetically depends on the aggregation speed; the cells with high affinity have the faster cellular aggregates). Cell subpopulations exhibiting strong affinity could aggregate more rapidly compared to other subpopulations. Thus, during the pdGC cultivation, we filtered the cellular aggregates by time once the cellular aggregates were formed.

**How long between the 3 screens?

Our response: The 3rd screened pdGC subpopulation were used right after filtration for 2nd screened pdGC subpopulation.

Regarding my comment:

"Line 331: cells from the 3rd screen (external type) express pathways linked to ecm organization, cell-cell/matrix adhesion which are typical of more differentiated cells. This is in apparent contradiction to data from figure 3e where cells from the internal types express these pathways."

And the author's response: For investigating the intratumor heterogeneity, we collected the subpopulation from single pdGC by filtering the pdGC to separate and screen the bigger sized pdGC subpopulation (>70-μm micropores), that presumably possesses high cell affinity within the subpopulation. We repeat the filtration process to collect the second (2nd)-screened and the third (3rd)-screened, so we aimed to compare different subpopulation within a single tumor line based on cell affinity. As presented in Fig. 5f, we would like to suggest what difference is found from TS15-88_3rd cell population, which showed significant changes in TMZ response, comparing to the rest of subpopulation groups (TS15-88 1st, 2nd, TS14-48 1st, 2nd, and 3rd). Indeed, Gene Ontology (GO) analysis revealed that TS15-88_3rd cells exhibited marked upregulation of ECM remodeling-related biological process, as well as the canonical Wnt signaling pathway, comparing to the rest of subpopulation groups. In addition, ECM-related cellular components such as collagen-containing extracellular matrix and integrin complex (its binding proteins) are significantly upregulated in TS15-88_3rdsubpopulation. In addition, as described in Fig. 5h, the gene set enrichment analysis (GSEA) also showed upregulated KEGG_ECM receptor interaction (NES=2.985, FDR=0.244), while resulted in downregulated trend in KEGG_cell adhesion molecules (CAMs) (NES=-2.159, FDR=0.587). With the GSEA result of KEGG_cell adhesion molecules (CAMs), we initially intended to show the similar trends of the rest of subpopulation groups with internal types of pdGCs as we analyzed in our intertumor heterogeneity (shown in Fig. 3e). However, when considering our FDR threshold = 0.25, this trend is less significant. So, it would be better to remove this GSEA panel here, and display KEGG_ECM receptor interaction only to show the consistency with GO analysis.

**The authors did not adress the apparent contradiction that I pointed out.

 Our response: In addition to our prior response, we included GSEA data to provide further explanation. Figure 3e depicts enrichment score plots for the internal and external subgroups' different KEGG phenotypes. The internal type of pdGCs exhibited elevated KEGG pathways associated with cell adhesion molecules and focal adhesion, as previously mentioned. Assuming that the spatial arrangement is governed by cell affinity, the overexpression of pathways involving cell adhesion molecules in internal type of pdGCs makes sense, showing that the internal type of pdGCs has a stronger cell affinity than the external type.

The GO biological process "homotypic cell-cell adhesion" was downregulated in the third subpopulation of TS15-88, but elevated in the other subpopulations (NES=-3.501, FDR=0.038). (Fig. R2) It indicates that the third subpopulation of TS15-88 has a weak cell-cell affinity compared to the other groups, which is consistent with the findings in Figure 3e.

GO analysis and enrichment score plots given in Fig. 5f, and Fig. 5h demonstrated that the TS15-88 3rd cell population displayed a significant overexpression of ECM remodeling-related biological process, cellular component, molecular activities, and Reactome, compared to the other subpopulation groupings (TS15-88 1st, 2nd, TS14-48 1st, 2nd, and 3rd). The GO cellular component "extracellular matrix" is significantly increased (NES=3.749, FDR=0.009), the overexpression of KEGG ECM receptor interaction (NES=2.985, FDR=0.244) is strongly connected with the ECM remodeling-related pathway in the TS15-88 3rd subpopulation. It is well-known that ECM offers structural and biochemical support in malignancies to govern proliferation, self-renewal, and differentiation of stem cell-like subpopulations, in addition to chemoresistance [5, 6]. Therefore, we presume that the third subset of TS15-88 cells with low cell affinity exhibits an increased ECM remodeling mechanism that likely contributes to malignancy.

Since CAMs include immunoglobulin superfamily [IgCAMs], cadherins, and integrins, it is plausible that the GSEA results regarding KEGG cell adhesion molecules (CAMs) resulted in a downregulated trend (lower cell affinity in TS15-88 3rd and higher cell affinity in the rest of the subpopulations, which is consistent with Fig. 3e), but with less statistical significance (NES=-2.159, FDR=0.587).

To demonstrate the consistency of this study, it would be preferable to delete this GSEA panel, present KEGG ECM receptor interaction, and add GO homotypic cell-cell adhesion, as described previously.

Figure R2. Gene set enrichment analysis to show the differences between 3rd screened TS15-88 subpopulation and the rest subpopulation groups including TS15-88 1st, 2nd and TS14-48 1st, 2nd, 3rd). Gene set enrichment score plots for the genes regarding (a) KEGG pathway and (b) Gene Ontology (GO) analysis.

References

[1] Townes P. L., Holtfreter J. (1955). Directed movements and selective adhesion of embryonic amphibian cells. J. Exp. Zool. 128, 53-120. https://doi.org/10.1002/jez.1401280105

[2] Fagotto F. (2014) The cellular basis of tissue separation. Development. 141 (17): 3303–3318. https://doi.org/10.1242/dev.090332

[3] Steinberg M. S., Takeichi M. (1994). Experimental specification of cell sorting, tissue spreading, and specific spatial patterning by quantitative differences in cadherin expression. Proc. Natl. Acad. Sci. USA 91, 206-209. https://doi.org/10.1073/pnas.91.1.206

[4] Toda S., Blauch L. R., Tang S. K. Y., Morsut L., Lim W. A., (2018). Programming self-organizing multicellular structures with synthetic cell-cell signaling. Science. 361(6398): 156-162. https://doi.org/10.1126/science.aat0271

[5] Nallanthighal S., Heiserman J. P., Cheon D. J. (2019). The Role of the Extracellular Matrix in Cancer Stemness. Front Cell Dev Biol. 7: 86. https://doi.org/10.3389/fcell.2019.00086

Round 3

Reviewer 3 Report

Add the following information in the manuscript, if it's not there already:

The sorting process is based on the size of cellular aggregates, which hypothetically depends on the aggregation speed; the cells with high affinity have the faster cellular aggregates). Cell subpopulations exhibiting strong affinity could aggregate more rapidly compared to other subpopulations. Thus, during the pdGC cultivation, we filtered the cellular aggregates by time once the cellular aggregates were formed. The 3rd screened pdGC subpopulation were used right after filtration for 2nd screened pdGC subpopulation.

Regarding authors' comments: "To demonstrate the consistency of this study, it would be preferable to delete this GSEA panel, present KEGG ECM receptor interaction, and add GO homotypic cell-cell adhesion, as described previously."

Sure, plus add in the discussion the info about CAMs having different roles in stemness, malignancy etc as you presented it in your response, with minor spell check and editing

The other elements have been adressed

Author Response

Add the following information in the manuscript, if it's not there already:

The sorting process is based on the size of cellular aggregates, which hypothetically depends on the aggregation speed; the cells with high affinity have the faster cellular aggregates). Cell subpopulations exhibiting strong affinity could aggregate more rapidly compared to other subpopulations. Thus, during the pdGC cultivation, we filtered the cellular aggregates by time once the cellular aggregates were formed. The 3rd screened pdGC subpopulation were used right after filtration for 2nd screened pdGC subpopulation.

Our response: We elaborated the details about the process of the sorting from single pdGCs in the Results part, “3.4 Heterotypic cultivation of pdGC MAs to assess intratumoral heterogeneity”, which include the information above-mentioned.

Regarding authors' comments: "To demonstrate the consistency of this study, it would be preferable to delete this GSEA panel, present KEGG ECM receptor interaction, and add GO homotypic cell-cell adhesion, as described previously."

Sure, plus add in the discussion the info about CAMs having different roles in stemness, malignancy etc as you presented it in your response, with minor spell check and editing

Our response: We supplemented the explanation and added references about the different roles of CAMs in the discussion part, along with the introduction, of our revised manuscript.